# BPQP: A Differentiable Convex Optimization Framework for Efficient End-to-End Learning

**Jianming Pan**[1]*, **Zeqi Ye**[2]*,
**Xiao Yang**[3]†, **Xu Yang**[3], **Weiqing Liu**[3], **Lewen Wang**[3], **Jiang Bian**[3]
[1]University of California, Berkeley, [2]Nankai University
[3]Microsoft Research Asia
jianming_pan@berkeley.edu, liamyzq@gmail.com
{xiao.yang, xuyang1, weiqing.liu, lewen.wang, jiang.bian}@microsoft.com

## Abstract

Data-driven decision-making processes increasingly utilize end-to-end learnable deep neural networks to render final decisions. Sometimes, the output of the forward functions in certain layers is determined by the solutions to mathematical optimization problems, leading to the emergence of differentiable optimization layers that permit gradient back-propagation. However, real-world scenarios often involve large-scale datasets and numerous constraints, presenting significant challenges. Current methods for differentiating optimization problems typically rely on implicit differentiation, which necessitates costly computations on the Jacobian matrices, resulting in low efficiency. In this paper, we introduce BPQP, a differentiable convex optimization framework designed for efficient end-to-end learning. To enhance efficiency, we reformulate the backward pass as a simplified and decoupled quadratic programming problem by leveraging the structural properties of the Karush–Kuhn–Tucker (KKT) matrix. This reformulation enables the use of first-order optimization algorithms in calculating the backward pass gradients, allowing our framework to potentially utilize any state-of-the-art solver. As solver technologies evolve, BPQP can continuously adapt and improve its efficiency. Extensive experiments on both simulated and real-world datasets demonstrate that BPQP achieves a significant improvement in efficiency—typically an order of magnitude faster in overall execution time compared to other differentiable optimization layers. Our results not only highlight the efficiency gains of BPQP but also underscore its superiority over differential optimization layer baselines.

## 1 Introduction

In recent years, deep neural networks have increasingly been used to address data-driven decision-making problems and to generate final decisions for end-to-end learning tasks. Beyond explicit forward functions, some layers of the network may be characterized by behaviors of implicit outputs, such as the solutions to mathematical optimization problems, which can be described as differentiable optimization layers [1]. These can be treated as implicit functions where inputs are mapped to optimal solutions. In this manner, the network can incorporate useful inductive biases, including domain-specific knowledge, physical structures, and priors, thereby enabling more accurate and reliable decision-making. This approach has been integrated into deep declarative networks [2] for end-to-end learning and has proven effective in various applications, such as energy minimization [3, 4] and predict-then-optimize [5, 6] problems. Here, we focus on convex optimization due to

---

*Work done during an internship at Microsoft.
†Corresponding Author

38th Conference on Neural Information Processing Systems (NeurIPS 2024).

its broad applications in fields such as portfolio optimization [7], control systems [8], and signal processing [9], among others.

Optimization problems typically lack a general closed-form solution; therefore, calculating gradients for relevant parameters requires more sophisticated methods. These methods can be categorized based on whether an explicit computational graph is constructed, namely into explicit unrolling and implicit methods. Explicit methods [4, 10–12] involve unrolling the iterations of the optimization process, which incurs additional computational costs. Conversely, implicit methods leverage the Implicit Function Theorem [13] to derive gradients. Some of these methods [14, 1, 15] are tailored for specific problems, thus restricting the options for forward optimization and reducing efficiency. Alternatively, other approaches [2, 10] offer more general solutions for deriving gradients but face inefficiencies during the backward pass. There remains substantial potential for enhancing efficiency in these methodologies.

To enable rapid, tractable differentiation within convex optimization layers and further enhance the capabilities of the end-to-end learning paradigm, we propose a general, first-order differentiable convex optimization framework, which we refer to as the **B**ackward **P**ass as a **Q**uadratic **P**rogramming (BPQP). Specifically, BPQP simplify the backward pass for the parameters of the optimization layer, by reformulating first-order condition matrix into a simpler Quadratic Programming (QP) problem. This decouples the forward and backward passes and creates a framework that can leverage existing efficient solvers (with the first-order algorithm, Alternating Direction Method of Multipliers (ADMM) [16], as the default) that do not require differentiability in both passes. Simplifying and decoupling the backward pass significantly reduces the computational costs in both the forward and backward passes. This key idea is summarized in Fig. 1.

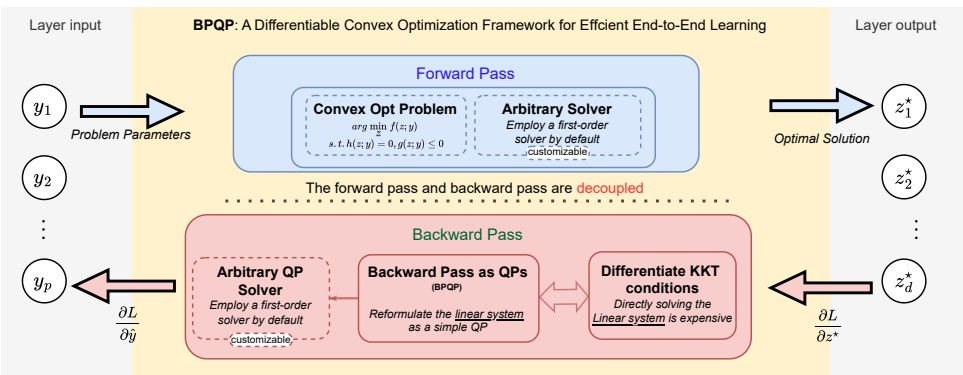

Figure 1: The learning process of BPQP: the previous layer outputs $y$ and generates the optimal solution $z^\star$ in the forward pass; the backward pass propagates the loss gradient for end-to-end learning; the process is accelerated by reformulating and simplifying the problem first and then adopting efficient solvers.

Our proposed framework has several theoretical and practical advantages:

- **Novel Backward Pass Reformulation:** Our framework BPQP decouples the backward pass from the forward pass of the convex optimization layer, then reforms the backward pass process to a simple QP problem. The method avoids solving the linear system involving the Karush-Kuhn-Tucker (KKT) matrix and enables *large-scale gradients computation* via ADMM. It leverages structural traits such as sparsity, solution polishing [16], and active-sets [17] for efficient and accurate gradients computation.

- **Efficient Gradients Computation:** Empirically, BPQP significantly improves the overall computational time, achieving up to $13.54\times$, $21.02\times$, and $1.67\times$ faster performance over existing differentiable layers on 100-dimension Linear Programming, Quadratic Programming, and Second Order Cone Programming, respectively. Such efficiency improvements pave the way for BPQP's application in large-scale real-world end-to-end learning scenarios, such as portfolio optimization. By adopting BPQP, the enhancement of the Sharpe ratio has increased from 0.65 (±0.25) to 1.28 (±0.43) compared to the widely-adopted two-stage approach methods.

- **Flexible Solver Choice:** BPQP accommodates any general-purpose convex solver to integrate the differentiable layer for end-to-end training. This flexibility in solver choice allows for better matching of solver capabilities with specific problem structures, potentially leading to improved efficiency and performance.

## 2  Related works

**Explicit methods**  Optimization problems typically do not have a general closed-form solution formula that expresses the decision variable in terms of other parameters. To address this challenge, explicit methods [4, 10, 11] unroll the iterations of the optimization process and use the decision variable from the final iteration as a proxy solution for the optimization problem. This constructs an explicit computational graph from the parameters to the proxy parameters, then calculate relevant gradients. Typically, these methods are designed for unconstrained optimizations. Applying them directly to constrained optimizations is computationally expensive because it requires projecting decision variables into a feasible region. Alt-Diff [12] is a novel unrolling solution that decouples constraints from the optimization and significantly reduces the computational cost. While advanced unrolling methods continue to improve their efficiency, they require an additional cost in the unrolled computational graph that increases with the number of optimization iterations.

**Implicit methods**  In contrast, implicit methods use the Implicit Function Theorem to relate the decision variable to other parameters. These methods specifically apply the theorem to KKT conditions in convex optimization. *Specificity-driven approaches* are often tailored for particular problems in convex optimization such as QP, offering good efficiency but sacrificing generality. OptNet [14] presented a differentiable batched-GPU QP solver. An ADMM layer is also developed for QP [18]. It facilitates implicit differentiation through a custom fixed-point mapping, reducing the cost by reducing the dimensions of the linear system to be resolved. *Generality-focused approaches* pursued more generalized solutions but faced limitations in efficiency, performing poorly on large-scale optimization problems. Their solution processes often rely on specific methods (such as coupled forward and backward passes), which prevent the application of various optimizers, thus leading to the aforementioned efficiency issues. Diffcp [1, 15] considers computing the derivative of a convex cone program by implicitly differentiating the residual map for its homogeneous self-dual embedding. Open-source convex solver CVXPY [19] adopts a similar method and computes gradients by SCS [20]. JaxOpt [10] proposes a simple approach to adding implicit differentiation on top of any existing solver, which significantly lowers the barrier to using implicit differentiation. Our work, BPQP, based on implicit methods, can be efficiently and broadly applied to convex optimization problems. We simplify the backward pass by reformulating it into a simpler, decoupled QP problem. This decoupling grants BPQP the freedom to choose an optimizer and, in conjunction with problem simplification, greatly reduces the computational cost in both the forward and backward passes. Notably, some work also examine discrete optimization challenges [21, 22], which are outside the purview of our study.

## 3  Background

### 3.1  Differentiable convex optimization layers

We will now provide some background information for the differentiable convex optimization layers.

Suppose the differentiable convex optimization layer has its input $y \in \mathbb{R}^p$, output $z^* \in \mathbb{R}^d$, we define the layer with the standard form of a convex optimization problem parameterized by $y$.

**Definition 1** (Differentiable Convex Optimization Layer)**.**  Given the input $y \in \mathbb{R}^p$, output $z^* \in \mathbb{R}^d$, a differentiable convex optimization layer is defined as

$$z^*(y) = \arg \min_{z \in \mathbb{R}^d} f_y(z)$$
$$\text{s.t. } h_y(z) = 0,$$
$$g_y(z) \leq 0,$$

where $y$ is the parameter vector of the objective functions and the constraints, $z$ is the optimization variable, $z^*$ is the optimal solution to the problem; The functions $f : \mathbb{R}^p \to \mathbb{R}$ and $g : \mathbb{R}^n \to \mathbb{R}$

are convex, and the function $h : \mathbb{R}^m \to \mathbb{R}$ is affine. The functions $f, h$, and $g$ are continuously differentiable *w.r.t* both $y$ and $z$, enabling the computation of $\frac{\partial z^*}{\partial y}$, .

The gradient of the parameter vector $y$ can be computed by combining the chain rule with the Implicit Function Theorem (IFT) [13]. Within the deep learning architecture, optimization layers are integrated alongside explicit layers within an end-to-end framework. Given the global loss function $\mathcal{L}$, the derivative of $\mathcal{L}$ *w.r.t* $y$ can be written as

$$\frac{\partial \mathcal{L}}{\partial y} = \frac{\partial \mathcal{L}}{\partial z^*} \frac{\partial z^*}{\partial y}.$$

We can easily obtain the derivative $\frac{\partial \mathcal{L}}{\partial z^*}$ through conventional automatic differentiation techniques applied to explicit layers. However, challenges arise in calculating $\frac{\partial z^*}{\partial y}$, the gradient of an implicit optimization layer. Considering the layer as an implicit function $\mathcal{F}(z^*, y) = 0$, recent studies such as OptNet [14] employ the IFT on the KKT conditions of the convex optimization problem to derive $\frac{\partial z^*}{\partial y}$. We state IFT here as a lemma for reference.

**Lemma 1** (Implicit Function Theorem). *Consider a continuously differentiable function $\mathcal{F}(z, y)$ with $\mathcal{F}(z^*, y) = 0$, and suppose the Jacobian matrix of $\mathcal{F}$ is invertible at a small neighborhood at $(z^*, y)$, we have*

$$\frac{\partial z^*}{\partial y} = -[\mathbf{J}_{\mathcal{F}}(z)]^{-1} \mathbf{J}_{\mathcal{F}}(y),$$

*where $\mathbf{J}_{\mathcal{F}}(z)$ and $\mathbf{J}_{\mathcal{F}}(y)$ are respectively the Jacobian matrix of $\mathcal{F}$ w.r.t $z$ and $y$.*

It should be emphasized that the differentiation of the KKT conditions requires calculating the optimal value $z^*$ in the forward pass and necessitates solving linear systems involving the Jacobian matrix in the backward pass. Both phases—forward and backward—are notably computationally intensive, particularly for large-scale problems. As a result, differentiating through KKT conditions directly does not scale efficiently to extensive optimization challenges.

## 3.2 Differentiating Through KKT Conditions

To differentiate KKT conditions more efficiently, CvxpyLayer [19] has adopted the LSQR technique to accelerate implicit differentiation for sparse optimization problems. However, this method may not be efficient for more general cases, which might not exhibit sparsity. Although OptNet [14] employs a primal-dual interior point method in the forward pass, making its backward pass relatively straightforward, it is suitable only for quadratic optimization problems.

In this paper, our main target is to develop a new method that can increase the computational speed of the differentiation procedure especially for general large-scale convex optimization problems.

We consider a general convex problem as defined in Definition 1. To compute the derivative of the solution $z^\star$ to parameter $y$, we follow the procedure of [14] to differentiate the KKT conditions using techniques from matrix differential calculus. Following this method, the Lagrangian is given by (omitting $y$),

$$L(z, \nu, \lambda) = f(z) + \nu^\top h(z) + \lambda^\top g(z), \tag{1}$$

where $\nu \in \mathbb{R}^m$ and $\lambda \in \mathbb{R}^n$, $\lambda \geq 0$ respectively denotes the dual variables on the equality and inequality constraints. The sufficient and necessary conditions for optimality of the convex optimization problem are KKT conditions. Applying the IFT (Lemma 1) to the KKT conditions and let $P(z^\star, \nu^\star, \lambda^\star) = \nabla^2 f(z^\star) + \nabla^2 h(z^\star)\nu^\star + \nabla^2 g(z^\star)\lambda^\star$, $A(z^\star) = \nabla h(z^\star)$ and $G(z^\star) = \nabla g(z^\star)$. Let $q(z^\star, \nu^\star, \lambda^\star) = \partial(\nabla f(z^\star) + \nabla h(z^\star)\nu^\star + \nabla g(z^\star)\lambda^\star)/\partial y$, $b(z^\star) = \partial h(z^\star)/\partial y$ and $c(z^\star, \lambda^\star) = \partial(D(\lambda^\star)g(z^\star))/\partial y$. Then the matrix form of the linear system can be written as:

$$\begin{bmatrix} P(z^\star, \nu^\star, \lambda^\star) & G(z^\star)^\top & A(z^\star)^\top \\ D(\lambda^\star)G(z^\star) & D(g(z^\star)) & 0 \\ A(z^\star) & 0 & 0 \end{bmatrix} \begin{bmatrix} \frac{\partial z^\star}{\partial y} \\ \frac{\partial \lambda^\star}{\partial y} \\ \frac{\partial \nu^\star}{\partial y} \end{bmatrix} = - \begin{bmatrix} q(z^\star, \nu^\star, \lambda^\star) \\ c(z^\star, \lambda^\star) \\ b(z^\star) \end{bmatrix}, \tag{2}$$

$D(\cdot) : \mathbb{R}^n \to \mathbb{R}^{n \times n}$ represents a diagonal matrix that formed from a vector and $z^\star, \nu^\star, \lambda^\star$ denotes the optimal primal and dual variables. Left-hand side is the KKT matrix of the original optimization

problem times the Jacobian matrix of primal and dual variables to $y$, e.g., $\frac{\partial z^\star}{\partial y} \in \mathbb{R}^{d \times p}$. Right-hand side is the negative partial derivatives of KKT conditions to the $y$.

We can then backpropagate losses by solving the linear system in Eq. (2). In practice, however, explicitly computing the actual Jacobian matrices $\frac{\partial z^\star}{\partial y}$ is not desirable due to space complexity; instead, [14] products previous pass gradient vectors $\frac{\partial \mathcal{L}}{\partial z^\star} \in \mathbb{R}^d$, to reform it by notations $[\tilde{z} \in \mathbb{R}^d, \tilde{\lambda} \in \mathbb{R}^m, \tilde{\nu} \in \mathbb{R}^n]$ (see Appendix A.3):

$$
\begin{bmatrix}
P(z^\star, \nu^\star, \lambda^\star) & G(z^\star)^\top & A(z^\star)^\top \\
D(\lambda^\star) G(z^\star) & D(g(z^\star)) & 0 \\
A(z^\star) & 0 & 0
\end{bmatrix}
\begin{bmatrix}
\tilde{z} \\
\tilde{\lambda} \\
\tilde{\nu}
\end{bmatrix}
= -
\begin{bmatrix}
(\frac{\partial \mathcal{L}}{\partial z^\star})^\top \\
0 \\
0
\end{bmatrix}.
\tag{3}
$$

And the direct gradients $\nabla_y \mathcal{L} \in \mathbb{R}^p = [q(z^\star, \nu^\star, \lambda^\star), c(z^\star, \lambda^\star), b(z^\star)][\tilde{z}, \tilde{\lambda}, \tilde{\nu}]^\top$.

## 4 Methodology

### 4.1 Backward Pass as QPs

Our approach solves Eq. (3) using reformulation. Consider a general class of QPs that have $d$ decision variables, $m$ equality constraints and $n$ inequality constraints:

$$
\underset{\tilde{z}}{\text{minimize}} \ \frac{1}{2} \tilde{z}^\top P' \tilde{z} + {q'}^\top \tilde{z} \quad s.t. \ A' \tilde{z} = b', \ G' \tilde{z} \leq c',
\tag{4}
$$

where $P' \in \mathbb{S}_+^d$, $q' \in \mathbb{R}^d$, $A' \in \mathbb{R}^{m \times d}$, $b' \in \mathbb{R}^m$, $G' \in \mathbb{R}^{n \times d}$ and $c' \in \mathbb{R}^n$. KKT conditions write down in matrix form:

$$
\begin{bmatrix}
P' & {G'}^\top & {A'}^\top \\
D(\tilde{\lambda}) G' & D(G' \tilde{z} - c) & 0 \\
A' & 0 & 0
\end{bmatrix}
\begin{bmatrix}
\tilde{z} \\
\tilde{\lambda} \\
\tilde{\nu}
\end{bmatrix}
=
\begin{bmatrix}
-q' \\
D(\tilde{\lambda}) c' \\
b'
\end{bmatrix}.
\tag{5}
$$

We note that Eq. (5) is equivalent to Eq. (3) if and only if: (i) $P' = P(z^\star, \nu^\star, \lambda^\star)$, $A' = A(z^\star)$, $D(\tilde{\lambda}) G' = D(\lambda^\star) G(z^\star)$, $[-q', D(\tilde{\lambda}) c', b'] = [-(\frac{\partial \mathcal{L}}{\partial z^\star})^\top, 0, 0]$ and (ii) $P(z^\star, \nu^\star, \lambda^\star)$ is positive semi-definite. However, $D(\tilde{\lambda}) G' = D(\lambda^\star) G(z^\star)$, which contains the unknown variable $\tilde{\lambda}$, may not hold. As the backward pass solves after the forward pass, we can change inequality constraints to an accurate active-set (i.e., a set of binding constraints) of equality conditions, and then condition (i) always holds for the equality-constrained QP. From this, the following theorem can be obtained (The detailed proof can be found in Appendix A.2)

**Theorem 1.** *Suppose that the convex optimization problem in Definition 1 is not primal infeasible and the corresponding Jacobian vector $\nabla_y \mathcal{L}$ exists. It is given by $\nabla_y \mathcal{L} = [q(z^\star, \nu^\star, \lambda^\star), c(z^\star, \lambda^\star), b(z^\star)][\tilde{z}, \tilde{\lambda}, \tilde{\nu}]^\top$ and $\tilde{z}, \tilde{\lambda}, \tilde{\nu}$ is the optimal solution of following equality constrained Quadratic Problem:*

$$
\underset{\tilde{z}}{\text{minimize}} \ \frac{1}{2} \tilde{z}^\top P' \tilde{z} + {q'}^\top \tilde{z} \quad s.t. \ A' \tilde{z} = b', \ G'_+ \tilde{z} = c'_+.
\tag{6}
$$

*Where $P' = P(z^\star, \nu^\star, \lambda^\star)$, $A' = A(z^\star)$, $G'_+ = G_+(z^\star)$ and $[-q', c'_+, b'] = [-(\frac{\partial \mathcal{L}}{\partial z^\star})^\top, 0, 0]$. $\lambda^\star$ and $\tilde{\lambda}$ only keep the elements according to the active-set after rewriting the inequality constraints to equality constraints. We did not create new notations for the sake of simplicity.*

Though our BPQP procedure described above also applies to Jacobians with forms other than vectors, e.g., matrices, in these cases where each 1-dimension column in $[\tilde{z}, \tilde{\lambda}, \tilde{\nu}]^\top$ right multiply the same KKT matrix and can be viewed as QPs packed in multi-dimensions, directly calculating the *inverse* of the KKT matrix may be more appropriate, especially when it contains a special structure like OptNet [14] and SATNet [23].

**General Gradients** The intuition of BPQP is that the linearity of IFT requires the KKT matrix left-multiply homogeneous linear partial derivative variables. Theorem 1 highlights a special situation that considers gradients at the optimal point (where KKT conditions are satisfied). Generally, BPQP

provides perspective to define gradients in parameter-solution space that preserves KKT norm. Let us consider a series of vectors denoting the $k$th iteration norm value of KKT conditions:

$$\|r^{(k)}\| = \| \left( r_{dual}^{(k)}, r_{cent}^{(k)}, r_{prim}^{(k)} \right) \| = C_k. \tag{7}$$

Where $r^{(k)} \in \mathbb{R}^{d+m+n}$ the KKT conditions in $k$th iteration and $C_k \in \mathbb{R}$ the norm value. The series $\{C_0, C_1, ..., C_k\}$ converges to 0 if the iteration algorithm is a contraction operator. Let $\mathcal{Q}^{(k)}$ denote standard QP problem w.r.t. parameter $P_k, q_k, A_k, b_k, G_k, c_k$ and decision variable $z_k$. At each iteration, BPQP yields $\nabla_y \mathcal{L}^{(k)}$ that preserves $\|r^{(k)}\| = C_k$. (See in Appendix A.4)

**Time Complexity** The time complexity of solving such QP is $\mathcal{O}(N^3)$ in the number of variables and constraints which is at the same level as directly solving the linear system Eq. (3). However, reformulation as QP provides substantial structures that can be exploited for efficiency, such that (we cover them in Section 4.2) sparse matrix, solution polishing [16], active-sets, and first-order methods, etc. Cleverly implementing BPQP, experiments at fairly large-scale dimensions in practice highlight BPQP's capacity in comparison to the state-of-art differentiable solver and NN-based optimization layers. Intuitively, BPQP is more efficient than previous methods because it utilizes the convex QP structural trait in the backward pass.

## 4.2 Efficiently Solve Backward Pass Problem with OSQP

The solver we referenced is OSQP [16], which incorporates the sparse matrix method and uses a first-order ADMM method to solve QPs, which we summarize below. On each iteration, it refines a solution from an initialization point for vectors $z^{(0)} \in \mathbb{R}^d$, $\lambda^{(0)} \in \mathbb{R}^m$, and $\nu^{(0)} \in \mathbb{R}^n$. And then iteratively computes the values for the $k + 1$th iterates by solving the following linear system:

$$\begin{bmatrix} P + \sigma I & A^\top \\ A & \mathrm{diag}(\rho)^{-1} \end{bmatrix} \begin{bmatrix} z^{(k+1)} \\ v^{(k+1)} \end{bmatrix} = \begin{bmatrix} \sigma z^{(k)} - q \\ \lambda^{(k)} - \mathrm{diag}(\rho)^{-1} \nu^{(k)} \end{bmatrix}, \tag{8}$$

And then performing the following updates:

$$\begin{aligned} \tilde{\lambda}^{(k+1)} &\leftarrow \lambda^{(k)} + \mathrm{diag}(\rho)^{-1} \left( v^{(k+1)} - \nu^{(k)} \right) \\ \lambda^{(k+1)} &\leftarrow \Pi \left( \tilde{\lambda}^{(k+1)} + \mathrm{diag}(\rho)^{-1} \nu^{(k)} \right) \qquad , \\ \nu^{(k+1)} &\leftarrow \nu^{(k)} + \mathrm{diag}(\rho) \left( \tilde{\lambda}^{(k+1)} - \lambda^{(k+1)} \right) \end{aligned} \tag{9}$$

where $\sigma \in \mathbb{R}_+$ and $\rho \in \mathbb{R}_+^n$ are the *step-size* parameters, and $\Pi : \mathbb{R}^m \to \mathbb{R}^m$ denotes the Euclidean projection onto constraints set. When the primal and dual residual vectors are small enough in norm after $k$th iterations, $z^{(k+1)}, \lambda^{(k+1)}$ and $\nu^{(k+1)}$ converges to exact solution $z^\star, \lambda^\star$ and $\nu^\star$.

In particular, given a backward pass problem Eq. (6) with known active constraints, as stated in OSQP, we form a KKT matrix below[3]:

$$\begin{bmatrix} P + \delta I & G_+^\top & A^\top \\ G_+ & -\delta I & 0 \\ A & 0 & -\delta I \end{bmatrix} \begin{bmatrix} \tilde{z} \\ \tilde{\lambda}_+ \\ \tilde{\nu} \end{bmatrix} = \begin{bmatrix} -q \\ 0 \\ 0 \end{bmatrix}, \tag{10}$$

As the original KKT matrix is not always invertible, e.g., if it has one or more redundant constraints, we modify it to be more robust for QPs of all kinds by adding a small regularization parameter $D(P + \delta I, -\delta I, -\delta I)$ (in Eq. (10)) as default $\delta \approx 10^{-6}$. We could then solve it with the aforementioned ADMM procedure to obtain a candidate solution, denoted as $\hat{t}$ and recover the exact solution $t$ from the perturbed KKT conditions $(K + \Delta K)\hat{t} = g$ by iteratively solving:

$$(K + \Delta K)\Delta \hat{t}^k = g - K\hat{t}^k. \tag{11}$$

where $\hat{t}^{k+1} = \hat{t}^k + \Delta \hat{t}^k$ and it converges to $t$ quickly in practice [16] for only one backward and one forward solve, which helps BPQP solve backward pass problems in a general but efficient way.

We have implemented BPQP with some of the use cases and have released it in the open-source library Qlib[24] (https://github.com/microsoft/qlib).

---

[3]$G_+ = G(z_+^\star)$ has the same row of active-set as $g(z_+^\star) = 0$, $z \in \mathbb{R}^{m+}$. $m_+$ is the number of active sets.

### 4.3 Examples: Differentiable QP and SOCP

Below we provide examples for differentiable QP and SOCP oracles (i.e. solutions) using BPQP. The general procedure is to first write down KKT matrix of the original decision making problem. And then apply Theorem 1. Assuming the optimal solution $z^\star$ is already obtained in forward pass.

**Differentiable QP** With a slight abuse of notation, given the standard QP problem with parameters $P, q, A, b, G, c$ as in Eq. (4). The result is exactly the same as OptNet [14] since both approaches are for accurate gradients. But BPQP is capable of efficiently solving large-scale QP forward-backward pass via ADMM [16], as shown in Section 5.1.

$$
\begin{aligned}
&\nabla_Q \mathcal{L} = \tfrac{1}{2}\left(\tilde{z}z^{\star T} + z^\star \tilde{z}^T\right) \quad \nabla_q \mathcal{L} = \tilde{z} && \nabla_A \mathcal{L} = \tilde{\nu}z^{\star T} + \nu^\star \tilde{z}^T \\
&\nabla_b \mathcal{L} = -\tilde{\nu} && \nabla_{G_+} \mathcal{L} = D(\lambda_+^\star)\tilde{\lambda}z^{\star T} + \lambda_+^\star \tilde{z}^T \quad \nabla_{c_+}\mathcal{L} = -D(\lambda_+^\star)\tilde{\lambda}
\end{aligned}
\tag{12}
$$

And $[\tilde{z}, \tilde{\nu}, \tilde{\lambda}]$ solves

$$
\operatorname*{minimize}_{\tilde{z}} \frac{1}{2}\tilde{z}^\top P \tilde{z} + \frac{\partial \mathcal{L}}{\partial z^\star}^\top \tilde{z} \quad \text{s.t. } A\tilde{z} = 0, \ G_+ \tilde{z} = 0.
\tag{13}
$$

**Differentiable SOCP** The second-order cone programming (SOCP) of our interest is the problem of robust linear program [25]:

$$
\operatorname*{minimize}_{z} q^\top z \quad \text{s.t. } a_i^\top z + \|z\|_2 \leq b_i \ i = 1, 2, ..., m.
\tag{14}
$$

where $q \in \mathbb{R}^d$, $a_i \in \mathbb{R}^d$, and $b_i \in \mathbb{R}$. With $m$ inequality constraints in $L_2$ norm, we give the gradients w.r.t. above parameters.

$$
\nabla_q \mathcal{L} = \tilde{z} \quad \nabla_{a_{i+}}\mathcal{L} = \lambda_{i+}^\star \tilde{z} + \lambda_{i+}^\star \tilde{\lambda}_i z^\star \quad \nabla_{c_{i+}}\mathcal{L} = \tilde{\lambda}_i, \ i = 1, 2, ..., m.
\tag{15}
$$

And $[\tilde{z}, \tilde{\nu}, \tilde{\lambda}]$ are given by ($t_1 = \sum_i \lambda_{i+}^\star$ and $t_0 = \|z^\star\|_2$)

$$
\operatorname*{minimize}_{\tilde{z}} \frac{1}{2}\tilde{z}^\top \left(\frac{t_1}{t_0}\mathbb{I} - \frac{t_1}{t_0^3}z^\star z^{\star\top}\right)\tilde{z} + \frac{\partial \mathcal{L}}{\partial z^\star}\tilde{z} \quad \text{s.t. } (a_{i+}^\top + \frac{1}{t_0}z^\star)^T \tilde{z} = 0, \ i = 1, 2, ..., m.
\tag{16}
$$

## 5 Experiments

In this section, we present several experimental results that highlight the capabilities of the BPQP. To be precise, we evaluate for (i) large-scale computational efficiency over existing solvers on random-generated constrained optimization problems including QP, LP, and SOCP, and (ii) performance on real-world end-to-end portfolio optimization task that is challenging for existing end-to-end learning approaches.

### 5.1 Simulated Large-scale Constrained Optimization

We randomly generate three datasets (e.g. simulated constrained optimization) for QPs, LPs, and SOCPs respectively. The datasets cover diverse scales of problems. The problem scale includes $10 \times 5, 50 \times 10, 100 \times 20, 500 \times 100$ (e.g., $10 \times 5$ represents the scale of 10 variables, 5 equality constraints, and 5 inequality constraints). Please refer to more experiment details in Appendix A.5.

**QPs Dataset** The format of generated QPs follows Eq. (6) to which the notations in the following descriptions align. We take $q$ as the learnable parameter to be differentiated and $\mathcal{L} = \mathbf{1}^\top z^\star$ in Eq. (3). To generate a positive semi-definite matrix $P$, $P'^\top P' + \delta I$ is assigned to $P$ where $P' \in \mathbb{R}^{d \times d}$ is a randomly generated dense matrix, $\delta I$ is a small regularization matrix, and $\delta = 10^{-6}$. Potentially, we set $c = Gz'$, $G \in \mathbb{R}^{m \times n}$, $z' \in \mathbb{R}^n$ to avoid large slackness values that lead to inaccurate results. All other random variables are drawn i.i.d. from standard normal distribution $N(0, 1)$.

**LPs Dataset** The LP problems are generated in the format below

$$
\operatorname*{minimize}_{z} \theta^T z + \epsilon \|z\|_2^2 \quad \text{s.t. } Az = b, Gz \leq h.
\tag{17}
$$

where $\theta \in \mathbb{R}^d$ is the learnable parameter to be differentiated, $z \in \mathbb{R}^d$, $A \in \mathbb{R}^{n \times d}$, $b \in \mathbb{R}^n$, $G \in \mathbb{R}^{m \times d}$, $h \in \mathbb{R}^m$ and $\epsilon \in \mathbb{R}_+$. All random variables are drawn from the same distribution as the QPs dataset.

It is noteworthy that it contains an extra item $\epsilon\|z\|_2^2$ compared with traditional LP. This item is added to make the optimal solution $z^\star$ differentiable with respect to $\theta$. Without this item, $P(z^\star, \nu^\star, \lambda^\star)$ is always zero and thus the left-hand side matrix becomes singular in Eq. (2). This is a trick adopted by previous work [7], and here we set $\epsilon = 10^{-6}$ as default.

**SOCPs Dataset** For SOCP in Eq. (14), we consider a specific simple case, i.e. $a_i = 0 \ \forall i$ and this relaxations results in $m = 1$. As in QP and LP, we take $q$ as differentiable parameter and set loss function $\mathcal{L} = \mathbf{1}^\top z^\star$, but all variables are drawn i.i.d. from standard Gaussian distribution $N(0, 1)$.

**Compared Methods** To demonstrate the effectiveness of BPQP, we evaluate the efficiency and accuracy of state-of-the-art differentiable convex optimizers, as well as **BPQP**, on the datasets mentioned above. The following methods are compared: **CVXPY** [15], **qpth/OptNet** [14], **Alt-Diff** [12], **JAXOpt** [10] and **Exact** [2]. Exact adopts the same algorithm as BPQP for the forward pass, but attempts to calculate exact gradients using direct matrix inversion on the KKT matrix during the backward pass.

**Evaluation and Metrics** To evaluate the efficiency of the compared methods, the runtime in seconds is used for each forward pass, backward pass, and total process. To evaluate the accuracy, we first get a target solution $z^{\text{Exact}}$ with a high-accuracy method and then calculate the cos similarity with compared methods (CosSim $= z^{\text{Exact}} \cdot z^{\text{method}_i}/(\|z^{\text{Exact}}\| * \|z^{\text{method}_i}\|)$). We ran each instance 200 times for average and standard deviation (marked in brackets) of the metrics.

Table 1: Efficiency evaluation of methods by runtime in seconds based on 200 runs, with lower numbers indicating better performance.

| dataset | metric | stage size method | Backward | | | | Total(Forward + Backward) | | | |
|---|---|---|---|---|---|---|---|---|---|---|
| | | | 10×5 | 50×10 | 100×20 | 500×100 | 10×5 | 50×10 | 100×20 | 500×100 |
| QP (small) | abs. time (scale 1.0e-04) | Exact | 37.2(±19.0) | 181.4(±55.2) | 460.2(±110.6) | 3027.1(±421.4) | 38.2(±18.8) | 187.9(±56.0) | 484.2 (±117.8) | 3495.5 (±446.1) |
| | | CVXPY | 43.1(±15.2) | 140.2(±22.5) | 627.8(±141.3) | 3054.8(±177.3) | 332.6(±77.2) | 615.4(±101.3) | 1862.9(±240.5) | 4716.2(±185.6) |
| | | qpth/OptNet | 26.3(±5.7) | 35.9(±8.9) | 41.3(±12.3) | **58.3(±26.5)** | 70.6(±13.3) | 770.8(±201.0) | 1030.2(±238.5) | 1872.8(±1254.0) |
| | | Alt-Diff | - | - | - | - | 150.5(±534.4) | 254.7(±72.1) | 475.3(±402.2) | 3044.7(±992.3) |
| | | BPQP | **0.6(±0.1)** | **2.8(±0.5)** | **11.0(±0.7)** | 104.2(±2.5) | **2.0(±0.5)** | **9.3(±2.9)** | **35.1(±3.8)** | **571.6(±130.7)** |
| | (scale 1.0e+00) | JAXOpt | 5.1(±2.4) | 8.0(±3.6) | 18.5(±8.2) | 383.7(±86.3) | 6.1(±3.0) | 12.4(±5.2) | 32.7(±13.6) | 729.8(±89.5) |
| LP | abs. time (scale 1.0e-04) | Exact | 39.1(±26.4) | 286.7(±103.7) | 595.3(±161.4) | 2656.0(±325.2) | 40.3(±26.4) | 290.3 (±104.1) | 620.0(±164.4) | 2949.1 (±378.5) |
| | | CVXPY | 39.3(±2.4) | 48.7(±5.1) | 72.5(±5.5) | 850.9(±45.2) | 291.6(±16.0) | 352.1(±36.2) | 930.4(±150.3) | 5046.1(±189.7) |
| | | qpth/OptNet | 25.2(±6.8) | 26.9(±6.2) | 28.6(±7.7) | 36.4(±0.9) | 854.7(±140.3) | 860.3(±190.6) | 963.1(±230.0) | 970.5(±252.5) |
| | | BPQP | **0.5(±0.1)** | **1.7(±0.2)** | **4.9(±0.4)** | **25.6(±7.5)** | **1.7(±0.1)** | **5.3(±1.6)** | **29.5(±6.7)** | **318.7(±106.4)** |
| SOCP | abs. time (scale 1.0e-04) | Exact | 2.3(±5.1) | 4.2(±6.3) | 12.6(±23.2) | 110.7(±116.8) | 47.5(±10.0) | 52.0(±8.4) | 73.3(±24.2) | 300.2 (±119.5) |
| | | CVXPY | 8.8(±0.9) | 8.9(±0.3) | 9.0(±0.6) | **11.1(±0.3)** | 64.1(±5.1) | 80.1(±3.4) | 105.0(±2.9) | 334.3(±3.2) |
| | | BPQP | **0.2(±0.0)** | **0.7(±0.0)** | **2.3(±0.0)** | 53.4(±0.3) | **45.4(±4.9)** | **48.5(±2.6)** | **63.1(±1.6)** | **242.9(±2.7)** |

Table 2: Large-scale comparison of efficiency evaluation of methods by runtime in seconds based on 10 runs, with lower numbers indicating better performance.

| dataset | metric | stage size method | Backward | | | | Total(Forward + Backward) | | | |
|---|---|---|---|---|---|---|---|---|---|---|
| | | | 500×200 | 1500×500 | 3000×1000 | 5000×2000 | 500×200 | 1500×500 | 3000×1000 | 5000×2000 |
| QP (large) | abs. time (scale 1.0e-01) | Exact | 4.1(±1.6) | 28.5(±8.7) | 86.2 (±14.0) | 249.0 (±17.3) | 4.5(±1.6) | 37.0 (±9.1) | 150.3 (±10.0) | 489.4 (±22.5) |
| | | Alt-Diff | - | - | - | - | 7.0(±1.3) | 42.5(±3.2) | 282.4(±152.8) | 1820.6(±59.4) |
| | | BPQP | **0.1(±0.0)** | **1.5(±0.1)** | **6.3(±0.5)** | **20.4(±0.4)** | **0.5(±0.0)** | **10.0(±0.8)** | **70.6(±4.7)** | **262.8(±17.0)** |

**Results** The results for efficiency evaluation are shown in Table 1. The evaluation covers three typical optimization problems with different problem scales. The results start from the QP dataset. Compared with state-of-the-art accurate methods, BPQP achieves tens to thousands of times of speedup in total time. We visualize part of the results of Table 1 in Figure 2, and perform sensitivity analysis under 500×100 setting in Figure 3, where the horizontal axis represents (tolerance, maximum iterations) of the methods. These results demonstrates the efficiency and robustness of BPQP. When the problem becomes large, such as 5000×2000, previous methods fail to generate results. CVXPY is extremely much slower because it reformulates the QP as a conic program and the reformulation is slow and has to be done repeatedly when the problem parameters change [16]. It is worth noting that BPQP is faster even in the backward pass, where CVXPY and qpth/OptNet share information from the forward pass to reduce computational costs. Sharing this information will limit the available forward solvers and result in a coupled design. Exact falls back to a simpler implementation that does not involve sharing information between designs. It solves the KKT matrix (i.e., Eq. (3)) in the backward pass via a matrix inverse method without relying on information from the forward pass. Although Exact uses a relatively efficient implementation in the forward pass (i.e., a first-order method, same as BPQP), the fallback backward implementation becomes a bottleneck for efficiency. The results of the LP dataset lead to similar conclusions as those of the QP dataset.

In the evaluation of the SOCP dataset, qpth/OptNet and Alt-Diff focus on QP and are excluded from this non-QP setting. Due to the specialty of SOCP, CVXPY does not require problem reformulation into conic programs, giving it an advantage. BPQP still outperforms other options in terms of total time across all problem scales.

Table 3: Backward accuracy of methods on simulated QP and non-QP(SOCP) dataset

| method | QP | | | | | SOCP | |
| | BPQP | CVXPY | qpth/OptNet | Alt-Diff | JAXOpt | BPQP | CVXPY |
| --- | --- | --- | --- | --- | --- | --- | --- |
| Avg. CosSim. | **0.992(±0.09)** | 0.924(±0.15) | 0.989(±0.12) | 0.985(±0.11) | 0.831(±0.14) | **1.00(±1.8e-013)** | 1.00(±1.3e-012) |

The accuracy evaluation results are shown in Table 1. In the forward pass, all solvers give nearly the same results, which are not shown in the table. When evaluating the backward accuracy, we use a matrix inverse method with high precision to solve Eq. (3) directly to get a target solution(i.e. $z^{Exact}$) and compare solutions from evaluated methods against it. The $CosSim.$ is relatively higher than that in the forward pass due to accumulated computational errors. Among them, the $CosSim.$ of our method BPQP is the highest in QP. The $CosSim.$ of all methods are small enough for SOCP.

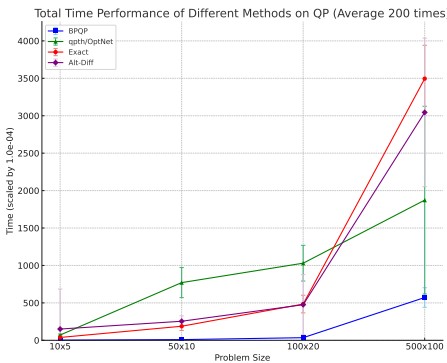

Figure 2: Table 1 results visualization.

Figure 3: Sensitivity analysis under 500×100 setting.

## 5.2 Real-world End-to-End Portfolio Optimization

Portfolio optimization is a fundamental problem for asset allocation in finance. It involves constructing and balancing the investment portfolio periodically to maximize profit and minimize risk. The problem is an important use case of end-to-end learning and can also be solved utilizing differentiable convex optimization layers [26]. We now show how to apply BPQP to the problem of end-to-end portfolio optimization (more experiment details in Appendix A.6).

**Mean-Variance Optimization (MVO)** [27] is a basic portfolio optimization model that maximizes risk-adjusted returns and requires long only and budget constraints.

$$\underset{w}{\text{maximize}}\ \mu^\top w - \frac{\gamma}{2} w^\top \Sigma w \ \text{ subject to } \mathbf{1}^\top w = 1,\ w \geq 0. \tag{18}$$

where variables $w \in \mathbb{R}^d$ represent the portfolio weight, $\gamma \in \mathbb{R} > 0$, the risk aversion coefficient, and $\mu \in \mathbb{R}^d$ the expected returns are the parameters to be predicted (under our setting, the input to the optimization layer) of the convex optimization problem. The covariance matrix, $\Sigma$, of all assets can be learned end-to-end by BPQP. However, it preserves a more stable characteristic than returns in time-series [28]. Therefore, we set it as a constant.

**Benchmarks** We evaluate BPQP based on the most widely used predictive baseline neural network, MLP. For the learning approach, we compared the separately two-stage(**Two-Stage**) and differentiable convex optimization layer approaches(**qpth/OptNet**). The optimization problem in the experiment has a variable scale of 500, which cannot be handled by other layers based on CVXPY and JAXOpt. We found the tolerance level for truncation in Alt-Diff hard to satisfy the 500 inequality constraints and yield a relatively longer training time (588 minutes per training epoch) than the above benchmarks. Our implementation substantially lowers the barrier to using convex optimization layers.

Table 4: Prediction and decision(portfolio) metrics evaluation of different methods in portfolio optimization. Speed is evaluated by training time per epoch (minute).

| | Prediction Metrics | | Portfolio Metrics | | Optimization Metrics | |
| --- | --- | --- | --- | --- | --- | --- |
| | IC ↑ | ICIR ↑ | Ann.Ret.(%) ↑ | Sharpe ↑ | Regret↓ | Speed↓ |
| Two-Stage | **0.033(±0.004)** | **0.32(±0.03)** | 9.28(±3.46) | 0.65(±0.25) | 0.0283(±0.0271) | **0.11** |
| qpth/OptNet | 0.026(±0.003) | 0.38(±0.12) | 16.54(±7.51) | 1.25(±0.42) | 0.0176(±0.0049) | 21.2 |
| BPQP | 0.026(±0.002) | 0.28(±0.03) | **17.67(±6.11)** | **1.28(±0.43)** | **0.0129(±0.0020)** | 7.7 |

**Results** The overall results are shown in Table 4. As we can see in the prediction metrics, Two-Stage performs best. Instead of minimizing multiple objectives without a non-competing guarantee, Two-Stage only focuses on minimizing the prediction error and thus avoids the trade-off between different objectives. However, achieving the best prediction performance does not equal the best decision performance. BPQP outperforms Two-Stage in all portfolio metrics, although its prediction performance is slightly compromised. qpth/OptNet shows comparable performance with BPQP. But the average training time of BPQP is 2.75x faster than OptNet. These experiments demonstrate the superiority of end-to-end learning, which minimizes the ultimate decision error, over separate two-stage learning.

# 6 Further discussion

In this section, we discuss the potential for BPQP to be applied in non-convex problems.

When addressing non-convex problems, we may encounter two challenges. Firstly, the solution is only a local minimum. Secondly, the solution represents only a proximate solution near a local minimum. If an effective non-convex method (e.g. Improved SVRG [29]) is employed in the forward pass, BPQP is still equipped to reformulate the backward pass as a QP. This is because our derivations and theoretical analysis are equally applicable to non-convex scenarios.

BPQP allows for the derivation of gradients that preserve the KKT norm, as elaborated in Section 4.1 under "General Gradients.", which means that when KKT norm is small, BPQP can derive a high quality gradient. Therefore, when a non-convex solver used in the forward pass successfully achieves a solution that is close to or even reaches a local or global minimum, BPQP can still compute well-behaved gradients effectively. This capability underscores the robustness of BPQP and adaptability in handling the complexities associated with non-convex optimization problems.

Additionally, many non-convex problems can be transformed into convex problems, making our approach applicable in a broader range of scenarios.

While its hard to perform experiments on non-convex problem due to the lack of baselines, we hope that future work can employ BPQP to do further analysis.

# 7 Conclusion

We have introduced a differentiable convex optimization framework for efficient end-to-end learning. Prior work in this area can be divided into explicit and implicit methods, based on the construction of an explicit computational graph.Explicit methods unroll the iterations of the optimization process, incurring additional costs. Conversely, implicit methods often struggle to achieve overall efficiency in both computing the optimal decision variable during the forward pass and solving the linear system involving KKT matrix during the backward pass. Our approach, BPQP, is grounded in implicit methods and simplify the backward pass by reformulating it into a simpler decoupled QP problem, which greatly reduces the computational cost in both the forward and backward passes. Extensive experiments on both simulated and real-world datasets have been conducted, demonstrating a considerable improvement in terms of efficiency.

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

# A Appendix

## A.1 Additional Related Works.

In this section, we discuss additional related works which focus on scenarios that regard optimization problem as terminal objectives. When optimization solutions solely constitute the terminal output of an end-to-end learning process—rather than intermediates within a neural network architecture—additional methodologies emerge.

### A.1.1 Learn-to-optimize

Learn-to-optimize has relatively low accuracy, which means it can only support some problems with a high error tolerance. DC3 [30] and ProjectNet [31] are research works in this direction. They leverage the universal approximation ability of neural networks and choose error correction algorithms to modify the output solution into the feasible region. [32] exploits energy-based model for decision-focused learning.

As a comparison to compute for exact gradients, existing work on *Learn-to-Optimize* trains an approximated solver network via SGD (e.g. DC3 [30]) or RL policy gradients [33–36] to solve constrained optimization problems that have a true graphical structure e.g. TSP, VRP, Minimum Vertex Cover, Max-Cut, and their variants. Leveraging the strong representation ability of state-of-the-art graph-based networks, RL obtains final solutions or intermediate results to be polished by searching or optimization algorithms. When optimizing convex and hard constraints in real-world scenarios, the underlying graph is typically fully connected and the accuracy tolerance is lower [37]. This presents a restriction on the widespread usage of works based on approximated solvers.

### A.1.2 Surrogate Loss

Alternatively, *surrogate loss* methods, arises from *predict-then-optimize* problem setting, present another avenue. LODL [38] introduces a generalizable surrogate loss, albeit at a high computational cost, limiting its widespread applicability. Some work[39] employ linear, low-dimensional representations of convex problems, yield approximate surrogate losses but at the expense of precision. Both SurCo [40] and SPO [6] explore linear surrogate models, yet their inability to capture the full complexity of original problems limits their effectiveness. LANCER [41] proposes a smooth, learnable landscape surrogate that extends to novel optimization challenges, though it may compromise accuracy. Our approach, BPQP, served as a differentiable layer which is capable of back-propagating exact gradients, offering greater flexibility and utility in practical applications.

## A.2 Proof of Theorem 1

In this section, we'll demonstrate the proof of Theorem 1.

*Proof.* After the forward pass, the active sets of the original set are known. Thus, the original optimization problem in can be reformulated as

$$z^\star = \arg\min_{z \in \mathbb{R}^d} f(z) \quad \text{subject to } h(z) = 0, \ g_+(z) = 0, \tag{19}$$

Where $y$ is omitted for simplifying notations, $g_+$ has the same row of the active set as the original inequality constraints $g_{\hat{y}}$.

Accordingly, the matrix form of the KKT conditions, as shown in 2, can be rewritten as follows:

$$\begin{bmatrix} P_+(z^\star, \nu^\star, \lambda_+^\star) & G_+(z^\star)^\top & A(z^\star)^\top \\ G_+(z^\star) & 0 & 0 \\ A(z^\star) & 0 & 0 \end{bmatrix} \begin{bmatrix} \frac{\partial z^\star}{\partial y} \\ \frac{\partial \lambda_+^\star}{\partial y} \\ \frac{\partial \nu^\star}{\partial y} \end{bmatrix} = - \begin{bmatrix} q_+(z^\star, \nu^\star, \lambda_+^\star) \\ c_+(z^\star) \\ b(z^\star) \end{bmatrix}, \tag{20}$$

where $\lambda_+^\star$ represents the Lagrangian multiplier for the equality constraints $g_+$, and it only contains the dimensions of the active-set of $g_y$. The following equations are modified accordingly:

$$P_+(z^\star, \nu^\star, \lambda_+^\star) = \nabla^2 f(z^\star) + \nabla^2 h(z^\star)\nu^\star + \nabla^2 g_+(z^\star)\lambda_+^\star$$
$$G_+(z^\star) = \nabla g_+(z^\star)$$
$$q_+(z^\star, \nu^\star, \lambda_+^\star) = \partial(\nabla f(z^\star) + \nabla h(z^\star)\nu^\star + \nabla g_+(z^\star)\lambda_+^\star)/\partial y$$
$$c_+(z^\star) = \partial(g_+(z^\star))/\partial y$$

The direct gradients become $\nabla_y \mathcal{L} \in \mathbb{R}^p = [q_+(z^\star, \nu^\star, \lambda_+^\star), c_+(z^\star), b(z^\star)][\tilde{z}, \tilde{\lambda}_+, \tilde{\nu}]^\top$. Equation 3 is then converted to:

$$\begin{bmatrix} P_+(z^\star, \nu^\star, \lambda_+^\star) & G_+(z^\star)^\top & A(z^\star)^\top \\ G_+(z^\star) & 0 & 0 \\ A(z^\star) & 0 & 0 \end{bmatrix} \begin{bmatrix} \tilde{z} \\ \tilde{\lambda}_+ \\ \tilde{\nu} \end{bmatrix} = - \begin{bmatrix} (\frac{\partial \mathcal{L}}{\partial z^\star})^\top \\ 0 \\ 0 \end{bmatrix}. \tag{21}$$

Finally, the linear Eq. 21 system is equal to the KKT condition of the QP in theorem 1 displayed below:

$$\begin{bmatrix} P' & G_+'^\top & A'^\top \\ G_+' & 0 & 0 \\ A' & 0 & 0 \end{bmatrix} \begin{bmatrix} \tilde{z} \\ \tilde{\lambda}_+ \\ \tilde{\nu} \end{bmatrix} = - \begin{bmatrix} q' \\ c_+' \\ b' \end{bmatrix}. \tag{22}$$

Please note that Theorem 1 does not create new notations for $\lambda_+^\star, \tilde{\lambda}_+$, but reuses $\lambda^\star, \tilde{\lambda}$ for simplicity.

$\square$

## A.3 Differentiate Through KKT Conditions Using the Implicit Function Theorem

In this section, we give a detailed discussion on Eq. (3). The sufficient and necessary conditions for optimality for are KKT conditions:

$$\nabla f(z^\star) + \nabla h(z^\star)\nu^\star + \nabla g(z^\star)\lambda^\star = 0$$
$$h(z^\star) = 0$$
$$D(\lambda^\star)(g(z^\star)) = 0 \tag{23}$$
$$\lambda^\star \geq 0,$$

Applying the Implicit Function Theorem to the KKT conditions and let $P(z^\star, \nu^\star, \lambda^\star) = \nabla^2 f(z^\star) + \nabla^2 h(z^\star)\nu^\star + \nabla^2 g(z^\star)\lambda^\star$, $A(z^\star) = \nabla h(z^\star)$ and $G(z^\star) = \nabla g(z^\star)$ yields to Eq. (2). We can then backpropagate losses by solving the linear system. In practice, however, explicitly computing the actual Jacobian matrices $\frac{\partial z^\star}{\partial y}$ is not desirable due to space complexity; instead, we product some previous pass gradient vectors $\frac{\partial \mathcal{L}}{\partial z^\star} \in \mathbb{R}^d$, to reform it by noting that

$$\nabla_y \mathcal{L} = \begin{bmatrix} \frac{\partial z^\star}{\partial y}, & \frac{\partial \lambda^\star}{\partial y}, & \frac{\partial \nu^\star}{\partial y} \end{bmatrix} \begin{bmatrix} (\frac{\partial \mathcal{L}}{\partial z^*})^\top \\ 0 \\ 0 \end{bmatrix}, \tag{24}$$

The first term of left hand side is the transposed solution of Eq. (2) and above can be reformulated as

$$\nabla_y \mathcal{L} = [q, \ c, \ b] \underbrace{\begin{bmatrix} P(z^\star, \nu^\star, \lambda^\star) & D(\lambda^\star)G(z^\star) & A(z^\star) \\ G(z^\star)^\top & D(g(x^\star)) & 0 \\ A(z^\star)^\top & 0 & 0 \end{bmatrix}^{-1} \begin{bmatrix} -(\frac{\partial \mathcal{L}}{\partial z^*})^\top \\ 0 \\ 0 \end{bmatrix}}_{\text{BPQP solution: } [\tilde{z}, \tilde{\lambda}, \tilde{\nu}]^\top}. \tag{25}$$

## A.4 Preserve KKT Norm Gradients

In a typical optimization algorithm, each stage of the iteration gives primal-dual conditions $r^{(k)}$, we follow the procedures of BPQP and solve the corresponding QP problem $\mathcal{Q}^{(k)}$ to define general gradients $\nabla_y \mathcal{L}^{(k)}$. The key difference here is that instead of using the optimal solution to derive BPQP, we plug in the intermediate points. By IFT,

$$dr^{(k)} = K^{(k)}[dz, d\lambda, d\nu]^\top + \frac{\partial r^{(k)}}{\partial y} dy = 0. \tag{26}$$

where $K^{(k)}$ is the Hessian matrix (KKT matrix) at points $(z_k, \lambda_k, \nu_k)$. The general gradients $\nabla_y \mathcal{L}^{(k)}$ is given by $dr^{(k)} = 0$ and therefore $\|r^{(k)}\| = C_k$ preserves KKT norm.

## A.5    Simulation Experiment

### Compared Methods

In Section 5.1, we randomly generate simulated constrained optimization datasets with uniform distributions and varying scales. We use these datasets to evaluate the efficiency and accuracy of state-of-the-art differentiable convex optimizers as well as BPQP. The methods of comparison briefly introduced previously are now detailed below:

**CVXPY** is a universal differentiable convex solver [19, 15, 1]. SCS [42, 43] solver is employed to accelerate the gradients calculation process.

**qpth/OptNet** qpth is a GPU-based differentiable optimizer, OptNet [14] is a differentiable neural network layer that wraps qpth as the internal optimizer.

**BPQP** is our proposed method. Its forward and backward passes are implemented in a decoupled way. It adopts the OSQP [16] as the forward pass solver. In the backward pass, it reformulates the backward pass as an equivalent simplified equality-constrained QP. OSQP is also adopted in the backward pass to solve the QP.

**Exact** uses the same forward pass solver as BPQP. The optimization algorithm used for the forward pass is the OSQP [16], which is a first-order optimization algorithm that does not share differential structure information. In the backward pass, without using reformulation via BPQP, the Eq. (3) are solved using the matrix inversion method like [2]. As a result, this approach fails to achieve overall efficiency.

**JAXOpt** [10] is an open-sourced optimization package that supports hardware accelerated, catchable training and differentiable backward pass. Optimization problem solutions can be differentiated with respect to their inputs either implicitly or via autodiff of unrolled algorithm iterations.

**Alt-Diff** [12] adopts ADMM in specializing in solving QP problems with exact solutions as well as gradients w.r.t. parameters.

### Hardware Setting

All results were obtained on an unloaded 16-core Intel(R) Xeon(R) CPU E5-2630 v3 @ 2.40GHz. qpth runs on an NVIDIA GeForce GTX TITAN X.

### Choice of Solvers of BPQP

BPQP decoupled the forward and backward pass and provides flexibility of choosing solvers. Normally, the first-order solver is greatly preferred when the problem scale becomes large and is also robust for small problem scale. Therefore, the first-order solver is a good enough default value, which is also employed by our framework and experiments.

## A.6    Portfolio Optimization Experiment

Statistical Risk Model (SRM) is used to generate the covariance matrix of MVO in Section 5.2. It takes the first 10 components with the largest eigenvalues by applying PCA on stock returns in the last 240 trading days. SRM shows the best performance of the traditional data-driven approach for learning latent risk factors.

### Dataset & Metrics

This section provides a more detailed introduction to the datasets and metrics used in the experiments described in Section 5.2. The dataset is from Qlib [24] and consists of 158 sequences, each containing OHLC-based time-series technical features [44] from 2008 to 2020 in daily frequency. Our experiment is conducted on CSI 500 universe which contains at most 500 different stocks each day.

For the predictive metrics, we evaluate IC (Information Coefficient) and ICIR (IC Information Ratio) of predictive model baselines. IC measures the correlation coefficient between the predicted stock returns $\hat{y}$ and the ground truth $y$. At each timestamp $t$, $IC^{(t)} = corr(\hat{y}^{(t)}, y^{(t)})$ in which

$$corr(\mathbf{x}, \mathbf{y}) = \frac{\sum_i (x_i - \bar{x})(y_i - \bar{y})}{\sqrt{\sum_i (x_i - \bar{x})^2 \sum_i (y_i - \bar{y})^2}}.$$

We report average IC across instances. $ICIR = \frac{mean(IC)}{std(IC)}$ measures both the average and stability of IC. A well-trained predictive model is expected to have higher IC and ICIR. For portfolio metrics, which measure the performance of investment strategies in the real market, we include two key indicators, $Ann.Ret.$ (Annualized Return) and $Sharpe$ (Sharpe Ratio) , which are the ultimate criteria widely used in quantitative investment. $Ann.Ret.$ indicates the return of given portfolios each year. $Sharpe = \frac{Ann.Ret.}{Ann.Vol.}$ in which $Ann.Vol.$ indicates the annualized volatility. To achieve higher $Sharpe$, portfolios are expected to maximize the total return and minimize the volatility of the daily returns. Transaction costs are not considered in our portfolio metrics to align with the regret loss and more stably demonstrate the effectiveness of end-to-end learning without being distracted by unconsidered random factors.

**Loss Construction**

The loss of Portfolio Optimization in end-to-end learning form can be constructed as

$$\mathcal{L}_{PO} = \underbrace{\mathbb{E}_{x,y\sim\mathcal{D}}\left[\left\|f_y\left(z^\star_{\hat{y}}\right) - f_y\left(z^\star_y\right)\right\|^2\right]}_{\text{Decision Error: Regret}} + \underbrace{\beta\mathbb{E}_{x,y\sim\mathcal{D}}\left[\|y - \hat{y}\|^2\right]}_{\text{Prediction Error}} + \alpha\underbrace{\mathcal{L}_{reg}(\theta)}_{\text{prior}},$$

where $x$ is the input data and $\theta$ is the parameter of the network. For we have ground truth values for the returns under this problem setting, we use the notation $\hat{y}$ to denote the input to the optimization layer, also the predicted returns, and $y$ for realized returns. On selecting $\beta$, we choose $\beta = \frac{\sigma_d^2}{\sigma_y^2} \in (0, 1)$ that denotes the ratio of variances between the random parameter $y$ and the decision error. Since the empirical regret suffers a more severe fluctuation over $y$ ($\sigma_d \gg \sigma_y > 0$) in convex optimization [5], prediction error should dominate in the end-to-end loss. However, we use an empiric distribution to approximate the Dirac distribution and set $\beta$ to a small ($\beta = 0.1$ in portfolio optimization experiment) but not zero value.

Under normality assumption, the MAP loss can be written as

$$\arg \max(p(\theta \mid \text{regret}, y, x)) \propto$$

$$\arg \max \prod_i \frac{1}{\sigma_d\sqrt{2\pi}} e^{-\frac{\text{regret}_i^2}{2\sigma_d^2}} \times \prod_j \frac{1}{\sigma_y\sqrt{2\pi}} e^{-\frac{(y_j - \hat{y}_j)_j^2}{2\sigma_y^2}}, \tag{27}$$

That is

$$\arg \min \sum_i \|f_y\left(z^\star_{\hat{y}}\right) - f_y\left(z^\star_y\right)^2 \|_i^2 + \frac{\sigma_d^2}{\sigma_y^2} \sum_j (y_j - \hat{y}_j)^2. \tag{28}$$

**Compared Methods**

Here is a more detailed explanation of the compared methods in this experiment.

**Two-Stage** separately learns a prediction MLP model to predict expected returns (i.e. $\mu$) and then generates decisions based on Eq. (18). All other methods below share the same prediction MLP model and only differ in the learning paradigm.

**qpth/OptNet** performs similar to BPQP, but with sightly lower performance in portfolio metrics and regret as it approaches exact gradient with a lower accuracy, shown in Table 3.

**BPQP** is our proposed method. All the accurate approaches (e.g. CVXPY, JAXOpt) have similar high-quality solutions in both forward and backward passes and are expected to have similar performance. Among them, only BPQP can efficiently handle the problem size of 500 variables(refer to Table 1), and thus BPQP are selected.

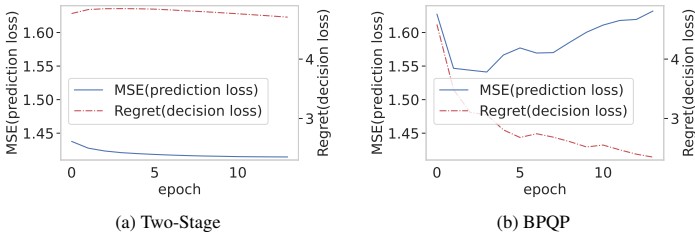

(a) Two-Stage          (b) BPQP

Figure 4: The prediction and decision error/loss of methods with different objectives

BPQP is trained using the loss function described in Section A.1.2.

To gain a deeper understanding of how end-to-end regret loss works, Figure 4 demonstrates the detailed learning curve of Two-Stage and BPQP. For each subfigure, the x-axis represents the number of epochs during training, and the y-axis represents the training loss of prediction and decision, respectively. Two-Stage aims to minimize the prediction loss, which is ultimately smaller than BPQP. However, the decision loss remains at a high level, resulting in a suboptimal decision. BPQP aims to minimize both prediction loss and decision loss. Both losses decrease initially, and then they start to compete in the later epochs. However, the decision error remains at a much lower value than Two-Stage, resulting in better decisions in the final evaluation.

### Experiment Setting

Here are the detailed search space for model architecture and hyper-parameters.

We use the same tolerance parameters for simulations experiments: Dual infeasibility tolerance: 1e-04, Primal infeasibility tolerance: 1e-04, Check termination interval: 25, Absolute tolerance: 1e-03, Relative tolerance: 1e-03, ADMM relaxation parameter: 1.6, Maximum number of iterations: 4000.

We use a lower tolerance parameter for real-world portfolio optimization experiments, due to the long-only strategy, we do not want small negative weight in the portfolio: Absolute tolerance: 1e-05, Relative tolerance: 1e-05, Dual infeasibility tolerance: 1e-05, Primal infeasibility tolerance: 1e-05.

**MLP predictor**: feature size: 153, hidden layer size: 256, number of layersr: 3, dropout rate: 0.Training: number of epoch: 30, learning rate: 1e-4, optimizer: Adam, frequency of rebalancing portfolio: 5 days, risk aversion coefficient: 1, early stopping rounds: 5, the inverse of beta (line 112): 0.1.

**DC3**: hidden size of solver net: 512, max stock size: 530, corrEps: 1e-4, corrTestMaxSteps: 10, softWeightEqFrac: 0.5, corrMomentum: 0.

### Additional experiments of approximate methods

In this section, we present additional experiments results for a typical learning-based approximate optimizer, DC3. DC3 are trained using the loss function described in Section A.1.2. We train the solver net (i.e. optimizer) with 500 epochs, 10000 samples, and 10 correction Test Max Steps for each type of QP and LP entries.

The computational cost scales linearly with the problem size and the number of model parameters. So, it is very efficient, especially when the scale is large. When the problem size is small (10×5 or 50×10), BPQP is still tens of times faster than DC3. However, DC3 becomes 5-10 times faster than BPQP when the problem size becomes large (500×100). Table 5 is the more detailed result of DC3 for both QP & LP (their problem size and number of model parameters are the same, so the time is nearly the same).

For large-scale real-world portfolio optimization, approximate methods such as DC3 can be a practical solution in terms of efficiency. The experiment results are shown in Table 6. Although DC3 is computationally efficient and thus applicable to large-scale real-world datasets, it performs poorly in decision metrics. This is due to the inaccurate gradient that deteriorates the learned model based on DC3. Therefore, accuracy is an important feature in end-to-end learning.

| size | 10×5 | 50×10 | 100×20 | 500×100 |
|---|---|---|---|---|
| forward + backward time(s) | 1.4e-02 | 1.5e-02 | 1.7e-02 | 1.7e-02 |

Table 5: Efficiency evaluation of the learn-to-optimize method DC3 by runtime in seconds.

Table 6: Prediction and decision(portfolio) metrics evaluation of DC3 in portfolio optimization. Speed is evaluated by training time per epoch (minute).

| | Prediction Metrics | | Portfolio Metrics | | Optimization Metrics |
|---|---|---|---|---|---|
| | IC ↑ | ICIR ↑ | Ann.Ret.(%) ↑ | Sharpe ↑ | Speed↓ |
| DC3 | 0.033(±0.001) | 0.31(±0.01) | -0.40(±0.97) | -0.16(±0.60) | 0.43 |

