# OpenReview forum: "BPQP: A Differentiable Convex Optimization Framework for Efficient End-to-End Learning"
_NeurIPS.cc/2024/Conference — NeurIPS 2024 spotlight_

### Official Review · Reviewer_Vrnz · 2024-07-12

**Soundness:** 3
**Presentation:** 3
**Contribution:** 2
**Rating:** 5
**Confidence:** 4

**Summary:**

The authors introduce BPQP, a differentiable convex optimization framework designed for efficient end-to-end learning. The core of this work lies in simplifying and decoupling the KKT matrix for the backward pass and solving it with a first-order solver to improve the overall efficiency of the module.

**Strengths:**

1. The paper is well-organized and well-written, making it relatively easy to read.

2. The proposed strategy performs well in the experiments.

**Weaknesses:**

1. The differences between BPQP and OptNet seem to be only reflected between Equation 3 and Equation 5. The rationale, necessity, and justification for this change may need more explanation.

2. For Equation 3, OptNet uses the interior-point method, while for Equation 5, BPQP uses ADMM. What is the motivation for using ADMM? The impact of using the interior-point method or various first-order/second-order optimization algorithms on performance (convergence, accuracy, time cost) also needs to be reflected in the experiments.

3. It is a good practice to learn from OptNet and conduct experiments on richer tasks such as Total Variation Denoising and Sudoku.

**Questions:**

See above.

---

> ### Author Rebuttal · Authors · 2024-08-01
>
> Thank you for your review. We will address each of the weaknesses and questions you have raised in detail.
>
> ## Q1&Q2. Differnece between BPQP and OptNet & The motivation of using ADMM
>
> > The differences between BPQP and OptNet seem to be only reflected between Equation 3 and Equation 5.
>
> > For Equation 3, OptNet uses the interior-point method, while for Equation 5, BPQP uses ADMM. What is the motivation for using ADMM? The impact of using the interior-point method or various first-order/second-order optimization algorithms on performance also needs to be reflected in the experiments.
>
> Thank you for your insightful questions regarding the distinctions between BPQP and OptNet (Amos & Kolter, 2017) and our choice of using ADMM in BPQP.
>
> The differentiation of the KKT matrix and the application of the Implicit Function Theorem (IFT) to these conditions are indeed standard steps in many convex optimization methods (as shown in Equations 1 and 2). Our primary contribution with BPQP, however, lies in our novel approach to the output of these processes. After applying the IFT to the KKT conditions, we reformulate the complex linear systems into simpler, more manageable quadratic problems (QPs) (illustrated in Equations 3, 4, 5, and Theorem 1). This reformulation enables the efficient computation of large-scale gradients and allows BPQP to completely decouple the backward pass from the forward pass. This decoupling provides significant flexibility in solver choice, enhancing the match between solver capabilities and specific problem structures, which potentially improves both efficiency and overall performance.
>
> Moreover, since BPQP fully decouples the backward pass from the forward pass, its performance is almost entirely dependent on the efficacy of the chosen solvers, both during the forward and backward passes. This decoupling underscores BPQP's adaptability and evolutionary capacity within a general framework for deriving backward pass gradients. We opted for the OSQP solver employing the ADMM method due to its strong performance in our framework. However, for other convex optimization challenges, methods like the interior-point or other first/second-order methods might be more suitable, and we remain open to using these alternatives. While we anticipate future work to explore replacing OSQP with more advanced solvers within the BPQP framework for potentially superior outcomes, conducting comparative experiments of different solvers within BPQP is not our current focus due to the vast array of choices and because it diverges from our primary research objectives.
>
> ## Q3. More real-world scenario experiments
>
> > It is a good practice to learn from OptNet and conduct experiments on richer tasks such as Total Variation Denoising and Sudoku.
>
> Thank you for your suggestion. OptNet indeed represents pioneering work in the field of differentiable convex optimization, and the use of tasks like Total Variation Denoising and Sudoku underscores its versatility in various real-world scenarios. However, these tasks typically involve relatively simple convex optimizations—such as small-scale QP, Linear Programming (LP), and Second Order Cone Programming (SOCP)—which we have already addressed through simulated experiments. These types of tasks require lower computational power and are less challenging, which means that, similar to OptNet, BPQP can easily handle them. However, such tasks do not fully demonstrate BPQP’s advantages when dealing with large-scale optimization problems.
>
> In contrast, our choice to focus on the portfolio optimization problem is deliberate. This problem is not only a quintessential example of convex optimization but also notably scalable, making it particularly suitable for transforming into the type of large-scale optimization problems that other convex optimization layers struggle with. This makes the portfolio optimization problem an excellent test case for evaluating the efficiency and accuracy of end-to-end learning methods like BPQP. Therefore, a comprehensive experiment on portfolio optimization adequately showcases BPQP's capabilities in efficiently deriving accurate results, thus aligning with our focus on demonstrating the framework's performance in more complex and large-scale scenarios.

---

> > ### Comment · Reviewer_Vrnz · 2024-08-10
> >
> > The author addressed some of my concerns. Although I am also inclined to accept this article, I do not intend to give a higher score at the moment.

---

> > > ### Author Response · Authors · 2024-08-10
> > >
> > > Thank you for reviewing our rebuttal! We appreciate your acknowledgment, and if you have any further concerns or suggestions, please feel free to share them with us at any time.

---

### Official Review · Reviewer_3Ly2 · 2024-07-12

**Soundness:** 4
**Presentation:** 3
**Contribution:** 4
**Rating:** 8
**Confidence:** 4

**Summary:**

This paper develops a technique to use deep learning models to solve convex optimization problems that offers speedups and space benefits over the current state of the art. Rather than using conventional implicit layers to predict optimal solutions, the authors consider the Karush-Kuhn Tucker (KKT) conditions for optimality in a different light in order to avoid costly and large Jacobian matrix computations. Here, they formulate this backward pass as a quadratic program (QP), which can be efficiently solved using first derivatives and requires less space and time than differentiating through the system of equations resulting from the KKT conditions. The KKT conditions are still used to form the backward pass as a QP, but solving the large system of equations is avoided.

**Strengths:**

The BPQP framework, due to avoiding solving the linear system of equations introduced by the KKT optimality conditions, offers significant speedups versus existing differentiable layers for LP, QP, and SOC problems. This is a very relevant area of research as NN-based end-to-end optimization solutions are gaining a lot of attention for the extreme speed-ups over using conventional optimization solvers for convex programs.

The comparisons show very promising results against a wide variety of other solutions, both conventional (CVXPY) and NN-based (OptNet).

The reformulation of the backward pass into a format which has efficient solution algorithms rather than naively solving the linear system of equations introduced by the optimality conditions is simple but quite clever, and the results speak for themselves.

**Weaknesses:**

I would prefer if BPQP was defined in the abstract (also, shouldn't it either be "as a Quadratic Program" or "as Quadratic Programming" to be grammatically correct?)

There are quite a few grammatical errors in the paper. Not to the level of affecting readability, but the authors may want to go through and polish this.

A bit nitpicky, but ADMM is not a "solver", it is an algorithm (page 2).

The text in Figure 1 in the forward and backward pass is very small.

**Questions:**

During inference, if the input to the model is actually infeasible (no solution exists that satisfies the corresponding KKT conditions), is there any indication of infeasibility produced by the model?

Do the authors have an idea of how this would extend to nonconvex problems? The KKT conditions would provide local minima, but perhaps the same reformulation of the backward pass as a QP would be more complicated.

**Limitations:**

The authors have adequately addressed the limitations and I do not see any potential negative societal impacts.

---

> ### Author Rebuttal · Authors · 2024-08-01
>
> Thank you for your review. We will address each of the weaknesses and questions you have raised in detail.
>
> ## Q1. Infeasible input
>
> > During inference, if the input to the model is actually infeasible (no solution exists that satisfies the corresponding KKT conditions), is there any indication of infeasibility produced by the model?
>
> Thank you for raising this important issue. In situations where the predicted parameter $y$ influences the constraints of the optimization problem, there indeed exists the possibility that the constraints could become too restrictive, resulting in an empty feasible set.
>
> Alternatively, the objective function might become unbounded with certain values of $y$. We acknowledge the significance of this concern and appreciate your insight. We will contemplate strategies to address these potential scenarios.
>
> However, it is important to note that in most practical applications and real-world scenarios, the formulation of the problem typically involves $y$ appearing only within a well-defined objective function. The constraints of the optimization problem generally encompass fixed physical constraints that do not change based on $y$ (as seen in methods like Alt-diff (Sun et al., 2022) and OptNet (Amos & Kolter, 2017)). Under these common conditions, the issue of input infeasibility due to $y$ influencing the constraints is unlikely to occur.
> Once again, thank you for your comment. We will continue to explore and refine our approaches to ensure robustness across all possible inputs.
>
> ## Q2. Extension to non-convex problems
>
> > Do the authors have an idea of how this would extend to nonconvex problems? The KKT conditions would provide local minima, but perhaps the same reformulation of the backward pass as a QP would be more complicated.
>
> Thank you for your insightful question regarding non-convex problems. When dealing with such problems, the solution might only reach a stationary point even without necessarily representing the local minimum. While achieving the global minimum is contingent the properties of the objective function, our framework, BPQP, is designed to be robust even in non-convex scenarios that a solution near a local minimum is provided. When adopting a efficient non-convex optimization method (e.g. Improved SVRG proposed in AllenZhu-Hazan, 2016a) achieves an approximate local minimum (also a KKT point), the gradient derived by BPQP may still hold good properties.
>
> Our theoretical analysis and derivations ensure that BPQP can still reformulate the backward pass as a simple Quadratic Program (QP). Also, BPQP can maintain nice gradient properties across both convex and non-convex contexts, which are detailed in Section 4.1 under "General Gradients", that the backward pass solution can preserve the KKT norm.
>
> We recognize the importance of ongoing advancements in non-convex solvers, as improvements in these solvers will enhance our ability to tackle more complex problems effectively. We are optimistic that as these solvers evolve, they will expand BPQP's applicability and efficiency in handling non-convex challenges.
>
> ## Q3. Minor points
> > I would prefer if BPQP was defined in the abstract.
>
> > There are quite a few grammatical errors in the paper. Not to the level of affecting readability, but the authors may want to go through and polish this.
>
> > A bit nitpicky, but ADMM is not a "solver", it is an algorithm (page 2).
>
> > The text in Figure 1 in the forward and backward pass is very small.
>
> Thank you for your valuable feedback. We appreciate your attention to detail and will define BPQP clearly in the abstract, correct terminology around ADMM, and address grammatical errors throughout the paper. Additionally, we'll ensure that the text in Figure 1 is enlarged for better readability, please refer to Figure R1 in the pdf document as a refined version. Your insights are instrumental in enhancing the quality of our manuscript.

---

> > ### Comment · Reviewer_3Ly2 · 2024-08-09
> > **Response to rebuttal**
> >
> > Thank you to the authors for responding to my comments. I'm glad to hear they have thought of these issues arising, and even if infeasible inputs are unlikely, future frameworks should consider all possible inputs to be the most robust. I don't have any further questions or comments.

---

> > > ### Author Response · Authors · 2024-08-09
> > >
> > > Thank you for your feedback and encouraging comments. We appreciate your insight and we will continue to refine our strategy for handling infeasible inputs to enhance the robustness and applicability of our framework.
> > >
> > > Thank you once again for your valuable contributions to our work!

---

### Official Review · Reviewer_qYw9 · 2024-07-15

**Soundness:** 2
**Presentation:** 3
**Contribution:** 3
**Rating:** 7
**Confidence:** 3

**Summary:**

The paper provides a novel approach for handling differentiable optimization layers when their forward corresponds to the solution to a convex constrained optimization problem. The authors show that the gradient of such a layer corresponds to the solution of a QP, thus enabling a tractable backward through the use of QP solvers (in contrast to previous approaches which require solving a linear system). The authors evaluate their proposed algorithm on synthetic data tasks and a portfolio optimization problem. The results show a significant improvement in efficiency with respect to baselines.

**Strengths:**

The high-level story of the paper is easy to follow. The proposed idea is simple and neat. Unlike existing approaches for implicit layers, it allows for the forward and backward passes to be decoupled algorithmically, thus allowing for extra flexibility since off-the-shelf solvers can be used for solving both the forward and backward problems. The results show a significant improvement in efficiency with respect to baselines.

**Weaknesses:**

* Although the proposed method applies when the forward passes correspond to general convex constrained optimization problems, only QPs are considered in the experiments section. Does the proposed algorithm provide as significant a speedup when the considered problem is less well-behaved?
* The paper does not include any comments on the limitations of the proposed method or potential future works. Note that there is a difference between scoping (e.g. saying that the problem applies only to the convex setting) and identifying a limitation (e.g. saying that the gains of an approach are less significant as the dimension of the problem increases).
* Tables 1 and 2 should include performance measures instead of just the computational time. Even if these synthetic problems are relatively trivial to solve with high precision, it is important to report the quality of the solution to ensure a fair comparison across approaches.
* The paper does not include any ablation experiments. See Question 1 for a potential ablation.

Minor points
* The authors could do a better job at motivating the field in the introduction: in which practical context have differentiable optimization layers been used and how (concretely) have they proven more effective than alternative approaches?
* Figure 1 is too small.
* The main paper results are only presented through tables. Plots help illustrate the results much better. Moreover, they make it easier to identify trends (e.g. as the size of the problem changes) and make it more evident if confidence intervals overlap across methods.

**Questions:**

1. Would the proposed method work if the forward problem is solved approximately, thus yielding an imprecise estimate of the optimal solution and Lagrange multipliers? Note that this often happens in practice due to computational cost considerations (which are the main motivation of the submission). Did the authors consider a sensitivity analysis of the precision of the solution to the forward problem?
2. Can the proposed algorithm be extended to non-convex problems?

**Limitations:**

* The authors do not mention any limitations of their work.
* There is no discussion on future work.
* No broader impact statement is provided in the paper.

---

> ### Author Rebuttal · Authors · 2024-08-01
>
> Thank you for your review. We will address each of the weaknesses and questions you have raised in detail.
>
> ## Q1. Effectiveness in less well-behaved problems
>
> > Only QPs are considered in the experiments section. Does the proposed algorithm provide as significant a speedup when the considered problem is less well-behaved?
>
> While Quadratic Programming (QP) is featured prominently in our experiments, we have also explored Linear Programming (LP) and Second Order Cone Programming (SOCP) in our Simulated Constrained Optimization experiments. The results, detailed in Table 1, demonstrate BPQP’s efficiency and generalization across different convex optimization problems, affirming the framework's robustness and adaptability. For our larger-scale simulated experiments, we followed the settings used in Alt-diff, and focused solely on QP to ensure comparability. These specific results are detailed in Table 2.
>
> ## Q2. Performance metrics
>
> > Tables 1 and 2 should include performance measures instead of just the computational time.
>
> Thank you for your suggestion. While Tables 1 and 2 focused on computational time, we have included gradient accuracy metrics in Table 3. These results show that BPQP achieves the highest accuracy in deriving backward pass gradients for QP and SOCP.
>
> ## Q3. Extension to non-convex problems & Sensitivity analysis
>
> > Can the proposed algorithm be extended to non-convex problems?
>
> When addressing non-convex problems, we often encounter two specific challenges. Firstly, the solution might only reach a local minimum. Secondly, the solution may be near the local minimum but not strictly satisfy the KKT conditions, representing only a proximate solution near a KKT point. If an effective non-convex method (e.g. Improved SVRG proposed in AllenZhu-Hazan, 2016a) is employed in the forward pass, capable of partially handling the second issue (like reach the KKT point in high efficiency & accuracy), BPQP is still equipped to reformulate the backward pass as a simple QP, as BPQP's framework allows for the derivation of gradients that preserve the KKT norm. Our derivations and theoretical analysis are equally applicable to non-convex scenarios.
>
> While we cannot ensure that the forward pass solver always finds a global minimum or strictly satisfies the KKT conditions, we can explore scenarios where the solver outputs a solution close to a stationary point and assess the quality of the resulting gradient. This aspect aligns closely with sensitivity analysis. Therefore, we will conduct sensitivity analysis on convex optimization problems to simultaneously evaluate BPQP's performance in scenarios resembling near-local optima within nonconvex optimization contexts.
>
> > Would the proposed method work if the forward problem is solved approximately, thus yielding an imprecise estimate of the optimal solution and Lagrange multipliers? Did the authors consider a sensitivity analysis of the precision of the solution?
>
> Thank you for raising this important point. Indeed, in practical settings, we often implement early stopping. In light of the challenges associated with non-convex roblems, we conducted a sensitivity analysis to explore scenarios where the forward pass only reaches near KKT points. Specifically, we analyzed the sensitivity of BPQP, OptNet (Amos & Kolter, 2017), and CVXPY (Agrawal et al., 2019b) under settings involving 500-dimensional variables with 100 equality and 100 inequality constraints, adjusting algorithm tolerance and maximum iterations. Please refer to Figure R2 in the pdf document. The results indicate that even when the forward pass solution is approximate—merely near stationary points—BPQP maintains high accuracy in computing the backward pass gradients, demonstrating robustness and effectiveness. This is largely due to BPQP's capability to preserve the KKT norm during the computation of backward pass gradients. From this, we can infer that in non-convex optimization scenarios, if the forward pass solver yields a reasonably good solution, using BPQP to compute the backward pass gradient would likely result in favorable outcomes.
>
> ## Q4. Limitations
> > The paper does not include any comments on the limitations of the proposed method or potential future works.
>
> Thank you for highlighting this oversight. The performance and limitations of BPQP largely depend on the choice of solver, as the framework completely decouples the forward and backward pass processes. As indicated in Table 1, while the forward pass time dominates the total computing duration for large-scale problems, the backward pass—formulated as a straightforward QP problem—is relatively quick to solve, showcasing our method's strength. However, both the efficiency and accuracy of BPQP are still contingent on the solver's capabilities.
>
> With a solver adept at handling large-scale convex optimization, BPQP's performance could be significantly enhanced. Moreover, if a robust non-convex solver is employed, BPQP is capable of deriving high-quality gradients, as mentioned earlier. Moving forward, we hope to see more research integrating advanced solvers within the BPQP framework to test its efficacy across diverse problem settings. This will help in further elucidating the scope and scalability of our approach.
>
> ## Q5. Minor points
>
> Thank you for your valuable feedback in minor points. To better motivate the introduction of BPQP, we will clarify the efficiency and effectiveness challenges in large-scale portfolio optimization.
>
> We have also addressed the concern regarding the size of Figure 1 by simplifying its design and adjusting its dimensions for better clarity. Please refer to the revised Figure R1 in the PDF document.
>
> Additionally, to enhance the presentation of our results, we have visualized the solving time of different methods for 500 variables with 100 equality and inequality constraints. This visualization can be found in Figure R3 in the PDF document.

---

> > ### Comment · Reviewer_qYw9 · 2024-08-07
> > **Reply to rebuttal**
> >
> > Overall, I am satisfied with the author's responses. In particular, I appreciate the discussion on limitations/future work and the ablation carried out during the rebuttal.
> >
> > I am thus raising my score to a 7.
> >
> > Some minor follow-ups:
> >
> > ---
> > > we have also explored Linear Programming (LP) and Second Order Cone Programming (SOCP) in our Simulated Constrained Optimization experiments.
> >
> > The setting I have in mind is problems where the objective function is not quadratic (or linear), but still convex.
> >
> > ---
> >
> > > Thank you for highlighting this oversight. The performance and limitations of BPQP largely depend on the choice of solver, as the framework completely decouples the forward and backward pass processes. As indicated in Table 1, while the forward pass time dominates the total computing duration for large-scale problems, the backward pass—formulated as a straightforward QP problem—is relatively quick to solve, showcasing our method's strength. However, both the efficiency and accuracy of BPQP are still contingent on the solver's capabilities.
> >
> > This point makes sense. The authors might want to emphasize this explicitly in the paper.

---

> > > ### Author Response · Authors · 2024-08-08
> > >
> > > Thank you for your thoughtful review and constructive comments. We greatly appreciate your recognition of our discussion on limitations and future work, as well as the ablation studies carried out during the rebuttal. We are encouraged by your decision to raise your score.
> > >
> > > We will make sure to explicitly emphasize the dependency of BPQP’s performance on the choice of solver in the limitations and future work section of our manuscript as suggested. Additionally, we are eager to explore and include less well-behaved examples as what you proposed (non-quadratic or linear objectives) in our experiments to further demonstrate the versatility and robustness of BPQP across a broader range of convex optimization scenarios.
> > >
> > > Thank you once again for your valuable feedback which has significantly contributed to improving our work.

---

### Official Review · Reviewer_wmCS · 2024-07-15

**Soundness:** 2
**Presentation:** 2
**Contribution:** 3
**Rating:** 7
**Confidence:** 5

**Summary:**

This paper proposes BPQP, a differentiable convex optimization framework to perform efficient end-to-end learning on convex problems using KKT condition. Compared with existing OptNet baselines, BPQP achieves similar performance but much faster speed.

**Strengths:**

1. This paper proposed BPQP, which utilizes the linearity of IFT to speed up the differential layers in an end-to-end framework. The intuition is good, and the performance of BPQP is good (2.75x faster than OptNet).

2. The paper is clearly written, and the theory part is well-explained with detailed proofs.

**Weaknesses:**

1. The BPQP framework proposed in this paper is solely built on KKT conditions, which means BPQP can only be applied to convex end-to-end problems. However, most real-world decision-making problems are more complex and non-convex, making BPQP less applicable.

2. In the experiment section, the author only performed two experiments: one is synthetic, and the other one is the portfolio optimization problem. However, I am a little worried if only the portfolio problem is enough to showcase the effectiveness of the proposed method. Besides, the author only compared with OptNet and two-stage, ignoring the other baselines in this area.

3. Despite the clear mathematical theorem and proof of BPQP, many of the mathematical aspects are actually similar to the OptNet work, which makes the paper less novel.

**Questions:**

My main question is: Is the portfolio experiment shown in section 5.2 enough to showcase the performance of the proposed method, or do we need more experiments? For the baseline, are the two-stage and OptNet enough, or do we still need more baselines?

**Limitations:**

1. More experiments, or at least, more datasets should be included in the results section to showcase the performance of BPQP.
2. In the experiments, the author should also add more baselines to compare with.

---

> ### Author Rebuttal · Authors · 2024-08-01
>
> Thank you for your review. We will address each of the weaknesses and questions you have raised in detail.
>
> ## Q1. Extension to non-convex problems
>
> > most real-world decision-making problems are more complex and non-convex
>
> Thank you for highlighting this important limitation. While it is true that many real-world problems are non-convex and this poses significant challenges, BPQP provides a viable approach even in these contexts.
>
> When addressing non-convex problems, we may encounter two challenges. Firstly, the solution might only reach a local minimum. Secondly, the solution may be near a local minimum but not strictly satisfy the KKT conditions, representing only a proximate solution. If an effective non-convex method (e.g. Improved SVRG proposed in AllenZhu-Hazan, 2016a) is employed in the forward pass, capable of partially handling some the second issue (like reach the local minimum in high efficiency & accuracy), BPQP is still equipped to reformulate the backward pass as a Quadratic Program (QP). This is because our derivations and theoretical analysis are equally applicable to non-convex scenarios.
>
> BPQP's framework allows for the derivation of gradients that preserve the KKT norm, as elaborated in Section 4.1 under "General Gradients." , which means that when KKT norm is small, BPQP can derive a high quality gradient. Therefore, when a non-convex solver used in the forward pass successfully achieves a solution that is reasonably close to or even reaches a local or global minimum, BPQP can still compute well-behaved gradients effectively. This capability underscores BPQP's robustness and adaptability in handling the complexities associated with non-convex optimization problems.
>
> Additionally, many non-convex problems can be transformed into convex problems, making our approach applicable in a broader range of scenarios.
>
> ## Q2. Real world experiments
>
> > Is the portfolio experiment shown in section 5.2 enough to showcase the performance of the proposed method?
>
> The portfolio optimization problem selected for our experiments is a quintessential example in convex optimization. It is notably scalable and versatile, making it an excellent proxy for testing the efficiency and accuracy of end-to-end learning methods. Many real-world problems are fundamentally simple convex optimizations, such as QP, Linear Programming (LP) and Second Order Cone Programming (SOCP) we mentioned in this paper. Therefore, a comprehensive experiment on portfolio optimization is sufficient to demonstrate BPQP's capabilities in deriving a accurate result efficiently.
>
> > For the baseline, are the two-stage and OptNet enough, or do we still need more baselines?
>
> Regarding additional baselines, besides the two-stage and OptNet (Amos & Kolter, 2017) comparisons already discussed, we have evaluated other prominent methods such as CVXPY, JAXOpt (Blondel et al., 2021), and Alt-diff (Sun et al., 2022). These methods were unable to complete the task on the large-scale data of the portfolio problem (Section 5.2, Lines 312-318). Furthermore, we have explored the 'learn-to-optimize' approach, another method for addressing end-to-end learning issues. However, our results indicate that BPQP, along with the two-stage and OptNet methods, significantly outperforms the 'learn-to-optimize' approach DC3 (Donti et al., 2021), as detailed in Appendix A.6.
>
> ## Q3. Similarity to OptNet in mathematical aspects
>
> > many of the mathematical aspects are actually similar to the OptNet work
>
> The differentiation of the KKT matrix and the application of the Implicit Function Theorem (IFT) to these conditions are common steps in many convex optimization methods. However, our main contribution with BPQP lies in how we handle the output from these processes. Unlike conventional approaches, after applying the IFT to the KKT conditions, we reformulate the resulting complex linear systems into simpler, more manageable quadratic problems, which enables efficient large-scale gradients computation. Also, BPQP completely decouples the backward pass from the forward pass, leading to the flexibility in solver choice, which allows for bettermatching of solver capabilities with specific problem structures, potentially leading to improved efficiency and performance.

---

> > ### Comment · Reviewer_wmCS · 2024-08-09
> >
> > Thanks a lot for your comment. I agree with your response and I will change the rate to 7.

---

> > > ### Author Response · Authors · 2024-08-09
> > >
> > > Thank you for your comprehensive and insightful review. Your feedback has been incredibly helpful in further improving our work.

---

### Author Rebuttal · Authors · 2024-08-01

We would like to thank all reviewers for their constructive feedback. Below, we summarize the major concerns raised and provide our  explanations.

## Q1. Can the proposed algorithm be extended to non-convex problems?
BPQP can provide a viable approach even in non-convex scenarios.

When addressing non-convex problems, we may encounter two challenges. Firstly, the solution is only a local minimum. Secondly, the solution represents only a proximate solution near a local minimum. If an effective non-convex method (e.g. Improved SVRG proposed in AllenZhu-Hazan, 2016a) is employed in the forward pass, BPQP is still equipped to reformulate the backward pass as a Quadratic Program (QP). This is because our derivations and theoretical analysis are equally applicable to non-convex scenarios.

BPQP allows for the derivation of gradients that preserve the KKT norm, as elaborated in Section 4.1 under "General Gradients." , which means that when KKT norm is small, BPQP can derive a high quality gradient. Therefore, when a non-convex solver used in the forward pass successfully achieves a solution that is close to or even reaches a local or global minimum, BPQP can still compute well-behaved gradients effectively. This capability underscores BPQP's robustness and adaptability in handling the complexities associated with non-convex optimization problems.

Additionally, many non-convex problems can be transformed into convex problems, making our approach applicable in a broader range of scenarios.

While its hard to perform experiments on non-convex problem due to the lack of baselines, we hope that future work can employ BPQP to do further analysis.

In fact, the scenarios where the solver outputs a solution close to a stationary point aligns closely with sensitivity analysis. Therefore, we will conduct sensitivity analysis on convex optimization problems to simultaneously evaluate BPQP's performance in scenarios resembling near-local optima within nonconvex optimization contexts.

## Q2. Did the authors consider a sensitivity analysis of the precision of the solution?

Indeed, in practical settings, we often implement early stopping due to computational cost considerations. We conducted a sensitivity analysis to explore scenarios where the forward pass only reaches near stationary points. Specifically, we analyzed the sensitivity of BPQP, OptNet (Amos & Kolter, 2017), and CVXPY (Agrawal et al., 2019b) under settings involving 500-dimensional variables with 100 equality and 100 inequality constraints, adjusting algorithm tolerance and maximum iterations. Please refer to Figure R2 in the pdf document. The results indicate that even when the forward pass solution is approximate, BPQP maintains high accuracy in computing the backward pass gradients. This is due to BPQP's capability to preserve the KKT norm during the computation of backward pass gradients. From this, we can infer that in non-convex optimization scenarios, if the forward pass solver yields a reasonable solution, using BPQP to compute the backward pass gradient would likely result in favorable outcomes.

## Q3. What is the main difference between BPQP and OptNet, especially in mathematical aspect?

The differentiation of the KKT matrix and the application of the Implicit Function Theorem (IFT) to these conditions, highlighted in OptNet, are now becoming standard steps in similar methods (as shown in Equations 1 and 2). Our primary contribution with BPQP, however, lies in our novel approach to the output of these processes. After applying the IFT to the KKT conditions, we reformulate the complex linear systems into simpler QPs (illustrated in Equations 3, 4, 5, and Theorem 1). This reformulation enables the efficient computation of large-scale gradients and allows BPQP to completely decouple the backward pass from the forward pass. This decoupling provides flexibility in solver choice, enhancing the match between solver capabilities and specific problem structures, which potentially improves both efficiency and overall performance.

## Q4. Do we need more real-world scenarios and baseline methods?

The portfolio optimization problem selected for our experiments is a typical example in convex optimization. It is notably scalable, making it an excellent proxy for testing the efficiency and accuracy of end-to-end learning methods. Many real-world problems are fundamentally simple convex optimizations, such as QP, Linear Programming and Second Order Cone Programming we mentioned in this paper. Therefore, a comprehensive experiment on portfolio optimization is sufficient to demonstrate BPQP's capabilities in deriving a accurate result efficiently.

Regarding additional baselines, besides the two-stage and OptNet comparisons already discussed, we have evaluated other prominent methods such as CVXPY, JAXOpt (Blondel et al., 2021), and Alt-diff (Sun et al., 2022). These methods were hard to handle the large-scale  portfolio problem (Section 5.2, Lines 312-318). Furthermore, we have explored the 'learn-to-optimize' approach DC3 (Donti et al., 2021). However, our results indicate that BPQP, along with the two-stage and OptNet methods, significantly outperforms DC3, as detailed in Appendix A.6.

## List of Figures

In response to the reviewers' suggestions, we have conducted additional experiments and redesigned three figures, which are included in the pdf document:
- Figure R1 is the refined version of Figure 1 for clarity: it presents with more readable fonts.
- Figure R2 shows our Sensitivity Analysis. We evaluated the robustness and performance of BPQP, OptNet, and CVXPY in settings involving 500-dimensional variables with 100 equality and inequality constraints, adjusting algorithm tolerance and maximum iterations. The results highlight BPQP's robustness and effectiveness.
- Figure R3 visualizes the total computation time under the same setting in Figure R2. This visualization demonstrates the trend and underscores BPQP's superior performance.

---

> ### Author Response · Authors · 2024-08-13
>
> We sincerely thank all the reviewers and Chairs for their time, effort, and insightful feedback during the review process!

---

### Decision · Program_Chairs · 2024-09-25

**Decision:**

Accept (spotlight)

**Comment:**

There is consensus among the expert referees that the submission should be accepted. The issues raised in the initial reviews were addressed in the detailed rebuttal, which should be incorporated in the paper.